# Temperature Elevation during Semen Delivery Deteriorates Boar Sperm Quality by Promoting Apoptosis

**DOI:** 10.3390/ani13203203

**Published:** 2023-10-13

**Authors:** Junwei Li, Wenming Zhao, Jiaqiao Zhu, Shuaibiao Wang, Huiming Ju, Shufang Chen, Athina Basioura, Graça Ferreira-Dias, Zongping Liu

**Affiliations:** 1College of Veterinary Medicine, Yangzhou University, Yangzhou 225009, China; lijunwei@yzu.edu.cn (J.L.); zhaowenming2018@126.com (W.Z.); jqzhu1998@163.com (J.Z.); hmju@yzu.edu.cn (H.J.); 2Jiangsu Co-Innovation Center for Prevention and Control of Important Animal Infectious Diseases and Zoonoses, Yangzhou University, Yangzhou 225009, China; 3DanAg Agritech Consulting (Zhengzhou) Co., Ltd., Zhengzhou 450000, China; billwang@danagintl.com; 4Royal Veterinary College, London NW1 0TU, UK; 5Ningbo Academy of Agricultural Science, Ningbo 315040, China; jhynku@163.com; 6Department of Agriculture, School of Agricultural Sciences, University of Western Macedonia, 53100 Florina, Greece; abasioura@uowm.gr; 7CIISA—Centre for Interdisciplinary Research in Animal Health, Faculty of Veterinary Medicine, University of Lisbon, 1300-477 Lisbon, Portugal; gmlfdias@fmv.ulisboa.pt; 8Associate Laboratory for Animal and Veterinary Sciences (AL4AnimalS), 1300-477 Lisbon, Portugal

**Keywords:** boar sperm, semen delivery, temperature changes, apoptosis, heat shock proteins, AMPK

## Abstract

**Simple Summary:**

Boar semen of high quality is a key factor that influences the outcome of artificial insemination. Furthermore, semen delivery practices could exert a negative effect on boar sperm quality and functionality. Nonetheless, the specific effect of temperature changes during semen delivery and molecular responses of sperm to temperature variations have not been fully understood. In this context, the present study aimed to investigate how temperature changes during semen transport influence boar sperm quality and functionality. Our results showed that the temperature of boar semen increased from the initial 17 °C to the ambient temperature of 37 °C after 24 h of delivery using a well-sealed Styrofoam. Sperm quality and functionality were greatly damaged, which could not be recovered when the semen temperature went back to 17 °C. Beyond this fact, boar sperm experienced a significant conversion of early apoptosis to late apoptosis caused by an elevated temperature. During this process, boar sperm adjusted themselves to survive by up-regulating the levels of heat shock proteins and activating the key energy regulator, AMP-activated protein kinase. The findings help to understand how temperature variations influence boar sperm quality and functionality and how the sperm struggles to survive, providing theoretical and technical information to improve the delivery practice of boar semen.

**Abstract:**

Semen delivery practice is crucial to the efficiency of artificial insemination using high-quality boar sperm. The present study aimed to evaluate the effect of a common semen delivery method, a Styrofoam box, under elevated temperatures on boar sperm quality and functionality and to investigate the underlying molecular responses of sperm to the temperature rise. Three pooled semen samples from 10 Duroc boars (3 ejaculates per boar) were used in this study. Each pooled semen sample was divided into two aliquots. One aliquot was stored at a constant 17 °C as the control group. Another one was packaged in a well-sealed Styrofoam box and placed in an incubator at 37 °C for 24 h to simulate semen delivery on hot summer days and subsequently transferred to a refrigerator at 17 °C for 3 days. The semen temperature was continuously monitored. The semen temperature was 17 °C at 0 h of storage and reached 20 °C at 5 h, 30 °C at 14 h, and 37 °C at 24 h. For each time point, sperm quality and functionality, apoptotic changes, expression levels of phosphorylated AMPK, and heat shock proteins HSP70 and HSP90 were determined by CASA, flow cytometry, and Western blotting. The results showed that elevated temperature during delivery significantly deteriorated boar sperm quality and functionality after 14 h of delivery. Storage back to 17 °C did not recover sperm motility. An increased temperature during delivery apparently promoted the conversion of sperm early apoptosis to late apoptosis, showing a significant increase in the expression levels of Bax and Caspase 3. The levels of phosphorylated AMPK were greatly induced by the temperature rise to 20 °C during delivery but reduced thereafter. With the temperature elevation, expression levels of HSP70 and HSP90 were notably increased. Our results indicate that a temperature increase during semen delivery greatly damages sperm quality and functionality by promoting sperm apoptosis. HSP70 and HSP90 could participate in boar sperm resistance to temperature changes by being associated with AMPK activation and anti-apoptotic processes.

## 1. Introduction

The modernized porcine industry is making more efforts than ever to reduce costs and improve reproductive efficiency. The supply of high-quality boar semen is a key point that contributes to the full use of boars with the best genes and speeds up genetic improvements. Multiple factors are involved in the production of high-quality boar semen, such as boar selection, health management, housing conditions, nutrition, semen collection practice, semen contamination, and semen handling [1]. Among those factors, heat stress, induced by animal exposure to a hot, humid environment without being able to adjust body temperatures, has been intensively investigated and found to reduce sperm quality and fertility [2]. Moreover, temperature and vibration management during transport exerts an impact on boar semen quality [3,4]. As the housing conditions and semen production practice have been greatly improved in the modern swine farm, heat stress at a body level is becoming a minor concern, while the effect of semen delivery is receiving more attention.

Typically, boar semen doses are delivered several hours after collection and packaging using a well-sealed insulated Styrofoam box or a car-loaded refrigerator under controlled temperatures of 16–18 °C. Delivery with a Styrofoam box is the most common method due to its low cost; however, it may impose severe damage to sperm quality due to the possible fluctuation in semen temperature, especially on hot summer days. A direct effect of environmental temperature on post-ejaculated mature sperm has been reported. Exposure to a higher temperature increases sperm motion parameters but shortens their lifespan because of an insufficient energy supply and overproduction of reactive oxygen species (ROS) [5,6]. Significant damage in sperm DNA was observed when the boar semen temperature was beyond 54 °C, while sperm motion parameters declined rapidly to zero above 40 or 42 °C [7]. Even though the adverse effect of in vitro exposure to an elevated temperature on semen quality has been intensively reported, the underlying molecular events induced by an elevated temperature in mature sperm remain to be further explored.

The molecular response of germ cells to heat stress has been suggested to be associated with changes in the pro-apoptotic Bax, anti-apoptotic Bcl-2, cytochrome C, caspases, and heat shock proteins (HSPs) [8,9]. AMP-activated protein kinase (AMPK), as a key energy sensor, plays an important role in cell survival. In somatic cells, AMPK activation is demonstrated to be linked to caspase-dependent mitochondrial apoptosis, promoting cell apoptosis [10,11]. In boar sperm, AMPK is localized at the acrosome and in the midpiece of the flagellum [12], which participates in the cell survival and fertilization process [13]. A heat-treatment-mediated AMPK/mTOR pathway is involved in the autophagy of lung cancer cells, which attenuates heat-induced apoptosis [14]. When suffering from H_2_O_2_-induced oxidative stress, AMPK can be activated, and it promotes cardiomyocyte death, which can be inhibited by HSP70 via reducing ROS accumulation [15], while heat stress in HepG2 cells induces dephosphorylation of AMPKα, which promotes HSP70 expression [16]. A previous study showed that HSP90 and Cdc37 bind specifically to the kinase domain of LKB-1 (upstream kinase for AMPK), regulating the stability of LKB-1 [17]. Heat stress can induce HSP production in cells, which function in protein homeostasis and apoptosis inhibition [18]. Amongst the HSP family, HSP70 and HSP90 have been associated with boar sperm quality and fertility [19]. In addition, studies have reported that either Hsp90 or Hsp70 can bind and block the activation of apoptotic protease activating factor 1 (Apaf-1) and indirectly inhibit pro-caspase activation and apoptosis, as well as enhance abnormal cell survival [20]. The protective effect of AMPK activation and HSP expression has been reported, while their relationship or interactive role in boar sperm survival under heat stress conditions remains obscure. In this context, this study aimed to disclose the molecular response of boar sperm when semen doses were exposed to heat stress induced by a regular semen delivery practice in the summer.

## 2. Materials and Methods

### 2.1. Semen Handling and Experiment Design

Boar ejaculates were provided by a commercialized boar stud (Shanghai Sunsing Co., Ltd., Shanghai, China). The boars of 1~2 years old for conventional production of semen doses were chosen in this study. The selected boars were healthy, mature, and fertile, housed in the boar stud under environmentally controlled conditions, and were given commercial feed according to semen donor requirements. Thirty ejaculates in total (10 Duroc boars × 3 ejaculates per boar) were used in this study. A gloved-hand method was utilized to collect ejaculates, which were thereafter diluted to a concentration for AI using a long-term commercial extender. The produced semen doses were subsequently packaged following the conventional practice and delivered to Yangzhou University in approximately 30 h in a well-sealed Styrofoam box together with ice bags. During semen delivery, the semen temperature was surveilled using a digital monitor. Semen temperature during delivery was within 17–22 °C. After receiving semen doses, semen quality re-check was performed. Only those semen samples that meet the requirements (sperm motility > 70%, sperm viability > 70%, and abnormality < 15%) were selected for the experiments. Three replicates were performed. For each replicate, ten semen doses of 80 mL were used for experiments, from each of which 10 mL was taken out and mixed to create one pooled semen sample of 100 mL. Each pooled semen sample was divided into two aliquots. In order to simulate semen delivery conditions under high temperatures in summer, one aliquot of each semen sample at 17 °C was wrapped with thermal insulation film and placed in a Styrofoam box together with an ice pack. Thereafter, the Styrofoam box was placed in an incubator at 37 °C after sealing. Temperature surveillance was conducted by inserting a digital temperature monitor. Semen samples were stored for 0 h, 5 h, 14 h, and 24 h, and their temperature reached 17 °C, 20 °C, 30 °C and 37 °C, respectively. Thereafter, semen samples were transferred to a refrigerator at 17 °C for subsequent 96 h storage to determine if cooling back to 17 °C could recover total sperm and progressive motility. Another aliquot was constantly kept in the refrigerator at 17 °C as control group. The temperature management is shown in Figure 1. At each time point, one part of semen samples was taken out and diluted with ACROMAX PLUS extender (ZoitechLab, Madrid, Spain) to a concentration of 25 × 10^6^ sperm/mL for evaluation of sperm quality and functionality. Another part of semen samples was taken out and centrifuged twice (2400× *g*, 3 min, 17 °C) to separate sperm and their surrounding fluid. The harvested sperm samples were washed three times with PBS by centrifugation (2400× *g*, 3 min, 17 °C) and stored at −80 °C for Western blotting analysis and intracellular ATP, ADP, and AMP assays.

### 2.2. Assessment of Sperm Quality and Functionality

Sperm motion parameters were determined in terms of total and progressive motility. A computer-aided sperm analysis system (CASA, ISASV1^®^; Proiser R + D, Paterna, Spain) was employed for an objective measurement. Briefly, a Makler counting chamber (Sefi Medical Instruments, Haifa, Israel) pre-warmed to 38 °C was loaded with 5 µL semen sample in a concentration of 25 × 10^6^ sperm/mL. At least 400 sperm from four to five fields were captured. In order to correct errors in sperm tracking, manual analysis was conducted carefully. The percentage of total motile spermatozoa (with an average path velocity ≥ 20 µm/s) and the percentage of sperm exhibiting rapid and progressive movement (straight line velocity ≥ 40 µm/s) were recorded as total motility and progressive motility, respectively.

The determination of sperm viability, intracellular ROS production, mitochondrial membrane potential, and apoptotic levels was performed using a flow cytometer (CytoFLEX S, Beckman Coulter Inc., Brea, CA, USA).

The sperm viability was determined using a triple-fluorescence procedure as previously described [21]. Briefly, a cytometric tube with 3 μL H-42 (Hoechst 33342, B2261, Sigma, Shanghai, China, 0.05 mg/mL in PBS), 2 μL PI (propidium iodide, P3566, Thermofisher, Shanghai, China, 0.5 mg/mL in PBS), and 2 μL PNA-FITC (L7381, Sigma, 200 μg/mL in PBS) was loaded with 100 μL sperm sample (25 × 10^6^ sperm/mL) and thereafter incubated for 10 min at 37 °C in the dark. Before uploading the semen samples to flow cytometry, addition of 400 μL PBS was performed to dilute the semen samples to an appropriate concentration for analysis. The percentage of viable sperm population showing negative PI and negative PNA-FITC was considered as sperm viability. The results of acrosome membrane damage were expressed as the percentage of sperm population exhibiting negative PI and positive PNA-FITC.

The intracellular levels of ROS were assessed by staining viable sperm population using CM-H2DCFDA (C6827, Thermofisher) as described by Guthrie and Welch [22]. Briefly, a cytometric tube was loaded with 1.5 μL of H-42 (0.05 mg/mL in PBS), 1 μL of PI (0.5 mg/mL in PBS), 1 μL of CM-H2DCFDA (1 mM in DMSO), 50 μL semen sample (25 × 10^6^ sperm/mL) and 950 μL of PBS. For the positive control group, 1 μL of TBH (458139, Sigma, 70% in distilled water) was loaded additionally. The samples were placed in an incubator at 37 °C for 30 min in the dark before analysis. Data were expressed as fluorescence intensity of million viable sperm showing PI-negative and DCF-positive.

The mitochondrial membrane potential was measured by staining viable sperm population using Mitotracker Deep Red 633 (M22426, Thermofisher) as previously depicted by Li et al. (2023) [21]. Briefly, cytometric tubes with 3 μL of H-42 (0.05 mg/mL in PBS), 2 μL of PI (0.5 mg/mL in PBS), and 5 μL of Mitotracker (0.2 μM in PBS of a stock solution of 1 mM in DMSO) were loaded with 100 μL sperm sample (25 × 10^6^ sperm/mL) and placed in an incubator at 37 °C for 15 min. Before flow cytometry analysis, 400 μL PBS was loaded to dilute the semen samples to the best concentration for analysis. The results were expressed as the percentage of viable sperm population exhibiting negative PI and positive Mitotracker.

Sperm apoptotic levels were determined in terms of early and late apoptosis using a commercial kit (A211-02, Vazyme, Nanjing, China). Translocation of phosphatidylserine (PS) from an inner to an outer leaflet of the sperm membrane indicates early apoptotic changes [23]. As Annexin V binds to sperm surface-expressed PS, the detection of PS can correspond to the level of sperm with early apoptotic changes. Briefly, 20 µL semen samples (25 × 10^6^ sperm/mL) were centrifuged (300× *g*, 5 min, 4 °C) to obtain sperm samples. The harvested sperm samples were washed twice with pre-cooled PBS by centrifugation (300× *g*, 5 min, 4 °C) and re-suspended with 100 µL Annexin-binding buffer. After addition of 5 µL Annexin V-FITC (original solution) and 5 µL PI (0.5 mg/mL in PBS), sperm samples were well mixed and incubated in dark at room temperature (20–25 °C) for 10 min. Before flow cytometric analysis, 400 µL Annexin-binding buffer was added to sperm samples. The results were recorded as percentage of viable sperm with negative PI and positive Annexin V (early apoptosis) and positive PI and positive Annexin V (late apoptosis).

### 2.3. Determination of Intracellular ATP, ADP, and AMP Content

Intracellular ATP, ADP, and AMP levels were determined following the protocol described by Nguyen et al. [24]. Briefly, 100 µL sperm samples in a concentration of 20 × 10^6^ cells/mL were mixed with 1 µL phosphatase inhibitor cocktail (P002, NCM Biotech, Suzhou, China) and kept for 30 min on ice. Thereafter, tubes containing 900 µL of boiling buffer (50 mM Tricine, 10 mM MgSO_4_, 2 mM EDTA, pH 7.8) were firstly heated for 5 min at 95 °C and then heated for 10 min at 95 °C after the incorporation of sperm samples. Subsequently, the samples were kept on ice for 10 min and then centrifuged at 5000× *g* for 30 min at 4 °C (5810 R, Eppendorf, Hamburg, Germany). The harvested supernatant was used for assays. Sperm samples were split into three aliquots (100 µL for each) and respectively incubated with 25 µL of Buffer A (75 mM Tricine, 5 mM MgCl_2_, and 0.0125 mM KCl, pH 7.5), Buffer B [Buffer A + 0.1 mM phosphoenolpyruvate (P7002, Sigma-Aldrich, Shanghai, China) + 0.08 µg/µL of the pyruvate kinase (P1506, Sigma-Aldrich)], and Buffer C [Buffer B + 0.1 µg/µL of the adenylate (myo) kinase (M3003, Sigma-Aldrich)]. Tubes containing Buffer A and B were incubated at 30 °C for 30 min, while tubes containing Buffer C were incubated at 30 °C for 90 min. When the time was up, boiling at 95 °C for 3 min was applied to all three tubes to stop reactions and then kept on ice for ATP assays. ATP content in the sperm samples was determined by incubating with Buffer A. Incubating with Buffer B leads to the transformation of ADP to ATP with the pyruvate kinase as a catalytic agent, which was therefore used to determine the combined amount of ATP and ADP according to reaction 1. Incubating with Buffer C leads to the conversion of AMP into ADP according to reaction 2 and the conversion of ADP into ATP according to reaction 1, which, therefore, was used to measure the combined amount of ATP, ADP, and AMP. The ATP content was determined by utilizing a commercial kit (FLAA, Sigma), following the manufacturer’s instructions. The ADP and AMP contents were calculated based on the reaction formulas. The results are expressed as pM/L.
(1)$$ADP+Phosphoenolpyruvat\overset{pyruvate\;kinase}{\to }ATP+Pyruvate$$
(2)$$AMP+ATP\overset{adenylate\;kinase}{\to }2ADP$$

### 2.4. Western Blotting

Semen samples containing 1.2 × 10^8^ sperm cells were centrifuged at 2400× *g* for 3 min at 4 °C to harvest sperm samples. Thereafter, sperm samples were washed three times with PBS by centrifugation (2400× *g*, 3 min, 4 °C) and re-suspended with RIPA buffer containing 1% protease and phosphatase inhibitor cocktail (EDTA-Free, 100× in DMSO) for 10 min at 4 °C. The sperm samples were further lysed by ultrasonication (20 KHz, 750 W, operating at 30% power, six cycles for 5 s on and 5 s off). After 30 min of lysis at 4 °C, the sperm protein samples were obtained by centrifugation at 12,000× *g* for 10 min at 4 °C. A portion of the supernatant was taken for the determination of protein concentration, while the rest was mixed with 5 × SDS loading buffer and boiled at 95 °C for 10 min. The proteins were separated by 10% SDS-PAGE at 90 V for 120 min and subsequently transferred onto PVDF membranes at 220 mA for 90 min. Western blotting analyses were performed using antibody of phospho-Thr172 AMPKα (2535S, Cell Signaling Technology, Danvers, MA, USA, 1:1000), LKB-1 (ab15095, Abcam, Cambridge, UK, 1:500), CaMKK2 (sc-271674, Santa Cruz, Santa Cruz, CA, USA, 1:500), Bcl-2(sc-7382, Santa Cruz, 1:1000), Bax (14796S, Cell Signaling Technology, 1:1000), Caspase 3 (9662S, Cell Signaling Technology, 1:1000), HSP70 (M20033M, Abmart, Shanghai, China, 1:1000), HSP90 (M20032M, Abmart, 1:1000) and anti-α-tubulin (AF0001, Beyotime, Shanghai, China, 1:1000) as primary antibodies. An automatic chemiluminescence image analysis system (Tannon 5200) was used to detect the membrane signal. ImageJ software (1.50i, Image, Inc., New Orleans, LA, USA) was employed for analysis of protein quantity in terms of gray intensity.

### 2.5. Statistical Analysis

The data analysis was performed using IBM SPSS software (version 20.0). The Kolmogorov–Smirnov test was employed to test the normality of the data set based on the residuals. Unpaired *t*-test was used to compare the mean values between two groups. The general linear model was used to analyze interactions between storage time and treatment methods. The one-way or the Kruskal–Wallis one-way ANOVA was utilized where appropriate to conduct comparisons between three or more groups, followed by LSD multiple-comparison tests. The values of the parameters are presented as the means ± SEMs. A *p*-value < 0.05 indicates significant differences between groups.

## 3. Results

### 3.1. Effect of Temperature Changes during Delivery on Sperm Quality and Functionality

Both the delivery time and treatment methods influenced (*p* < 0.05) the total sperm and progressive motility. The interaction between the delivery time and treatment methods was significant for both sperm motility parameters assessed. As shown in Table 1, within 24 h of delivery, a decline (*p* < 0.05) of total sperm and progressive motility was observed in the treatment group (delivery under 37 °C) at 24 h compared to that at 0 h, while no significant changes were found in the control group (constantly stored at 17 °C). Total sperm and progressive motility in the control group were considerably higher (*p* < 0.05) than that in the treatment group at 24 h. At 24 h of delivery, sperm motility remained in the control group but decreased sharply to near zero in the treatment group. Thereafter, semen samples of the treatment group were transferred to a refrigerator at 17 °C to check if sperm quality could be restored. Recovery of total sperm and progressive motility in the treatment group was not observed. Total sperm and progressive motility in the control group continued to decrease (*p* < 0.05) after 48 h of storage compared to that in previous time points. After 96 h of storage, total sperm and progressive motility from both control and treatment groups were approximated to be zero.

Since a substantial decrease in the total sperm and progressive motility of the treatment group was observed after 24 h of delivery and no recovery of those parameters was displayed, sperm viability, acrosome integrity, intracellular ROS level, and mitochondrial membrane potential were determined exclusively within 24 h of delivery (Table 2). Both delivery time and treatment methods influenced (*p* < 0.05) those sperm parameters. The interaction between delivery time and treatment methods was significant for those sperm parameters assessed. In comparison with 0 h, sperm viability declined (*p* < 0.05) after 14 h of delivery in the control group, while a decrease (*p* < 0.05) was found in the treatment group after 5 h of delivery. Compared to the control group, the treatment group exhibited greater damage (*p* < 0.05) in sperm viability at 14 h and 24 h of delivery. An increase in acrosome damage was found in both the control and treatment groups, showing a higher (*p* < 0.05) value at 14 h than at 0 h of delivery in both groups. Compared with 0 h, the intracellular ROS level decreased (*p* < 0.05) in the control group, while no significant changes were found in the treatment group. With respect to the control group, the treatment group displayed a lower (*p* < 0.05) intracellular ROS level at 0 h and a higher (*p* < 0.05) level at 14 h of delivery. An increase in sperm mitochondrial membrane potential with the delivery time was found in both the control and treatment groups, being higher (*p* < 0.05) at 24 h than at 0 h of delivery in both groups.

### 3.2. Effect of Temperature Changes during Delivery and Subsequent Storage (at 17 °C) on Sperm Apoptotic Levels

In order to understand the damage in sperm quality and functionality induced by temperature changes during delivery, the early and late apoptotic levels of the sperm were determined during 24 h of delivery and subsequent storage at 17 °C for 3 days (Figure 1). Compared to 0 h, early apoptotic levels of sperm in the control group were maintained during 24 h of delivery but increased (*p* < 0.05) at 48 h and 96 h of storage, while no significant changes in the early apoptotic levels of the sperm in the treatment group were observed during delivery and subsequent storage. In contrast to the control group, the treatment group presented higher (*p* < 0.05) early apoptotic levels of the sperm at 5 h of delivery but lower (*p* < 0.05) levels at 48 h and 96 h of storage. In comparison to 0 h, the control group exhibited lower (*p* < 0.05) late apoptotic levels of sperm at 24 h of delivery and higher (*p* < 0.05) levels at 72 h of storage. The treatment group induced higher (*p* < 0.05) late apoptotic levels of sperm after 48 h of storage in comparison with that at 0 h. Compared with the control group, the treatment group showed higher (*p* < 0.05) late apoptotic levels of sperm at 5 h and 24 h of delivery and after 48 h of storage. During delivery and subsequent storage, the control group showed higher early apoptotic levels but lower late apoptotic levels, while the treatment group promoted conversion from early apoptosis to late apoptosis.

Expression levels of Bcl-2, Bax, and Caspase 3, key proteins in the mitochondria-mediated apoptotic pathway, were analyzed (Figure 2). During semen delivery at 24 h, the expression levels of Bcl-2 were not influenced in both the control and treatment groups. During subsequent storage, a reduction in Bcl-2 expression at 96 h (*p* < 0.05) in the control group and at 72 h (*p* < 0.05) and 96 h (*p* < 0.05) in the treatment group was observed when compared to that at 0 h. The treatment group induced higher (*p* < 0.01) levels of Bcl-2 than that in the control group at 96 h of subsequent storage. Expression levels of Bax increased (*p* < 0.05) with delivery or storage time in both the control and treatment groups. A significant increase in the levels of Bax in the control group was not observed during semen delivery of 24 h until subsequent storage for 72 h (*p* < 0.05) and 96 h (*p* < 0.01) in comparison with that at 0 h, while in the treatment group, expression levels of Bax increased at 14 h (*p* < 0.05) and 24 h (*p* < 0.05) of delivery, and continued to increase at 48 h (*p* < 0.05), 72 h (*p* < 0.01) and 96 h (*p* < 0.01) of subsequent storage when compared to that at 0 h. In contrast to the control group, the treatment group promoted the expression of Bax, especially at 5 h of delivery (*p* < 0.05). In the control group, no significant changes in expression levels of Caspase 3 were observed during delivery of 24 h, while subsequent storage for 48 h (*p* < 0.05) and 96 h (*p* < 0.01) induced higher levels of that when compared to 0 h. In the treatment group, an increase in Caspase 3 expression was found at 24 h (*p* < 0.01) of delivery and during subsequent storage for 48 h (*p* < 0.001), 72 h (*p* < 0.001), and 96 h (*p* < 0.001) when compared to 0 h. Compared to the control group, the treatment group promoted the expression of Caspase 3, especially at 72 h (*p* < 0.01) of subsequent storage.

### 3.3. Effect of Temperature Changes during Delivery and Subsequent Storage (at 17 °C) on Activation of Sperm AMPK and Expression Levels of HSP70 and HSP90

Delivery/storage duration and temperature changes influenced the activation of AMPK (Figure 3). Compared with 0 h, delivery of 5 h (*p* < 0.05) and 24 h (*p* < 0.05) and subsequent storage of 48 h (*p* < 0.05) exhibited higher levels of p-AMPK in the control group, while higher levels of p-AMPK were observed at delivery of 5 h (*p* < 0.01) and lower levels at subsequent storage of 48 h (*p* < 0.05) and 72 h (*p* < 0.05) in the treatment group. Compared to the control group, the treatment group showed lower levels of p-AMPK at subsequent storage of 48 h (*p* < 0.05) and 72 h (*p* < 0.05). The upstream kinases of AMPK, LKB-1, and CaMKK2 were determined, as well. The expression levels of LKB-1 were not affected by delivery or the subsequent storage time in both the control and treatment groups. In contrast, the treatment group exhibited higher levels (*p* < 0.05) of LKB-1 at subsequent storage of 96 h than that in the control group. Delivery and subsequent storage time did not influence the expression of CaMKK2 in the control group, while higher levels (*p* < 0.01) of that were observed at 5 h of delivery in the treatment group in comparison with that at 0 h. AMPK can be allosterically activated by AMP binding. Thus, the changes in the AMP/ATP ratio are directly involved in the phosphorylation of AMPK. The results showed that the AMP/ATP ratio increased at delivery of 5 h (*p* < 0.01) and 14 h (*p* < 0.001) and at subsequent storage of 48 h (*p* < 0.001), 72 h (*p* < 0.01) and 96 h (*p* < 0.001) in the control group when compared with that at 0 h. In the treatment group, the AMP/ATP ratio increased only at delivery of 24 h (*p* < 0.001) and subsequent storage of 96 h (*p* < 0.01) in comparison with that at 0 h. Compared to the control group, a decrease at delivery of 14 h (*p* < 0.05), an increase at delivery of 24 h (*p* < 0.05), and subsequent storage of 72 h (*p* < 0.05) in the AMP/ATP ratio were observed in the treatment group.

As shown in Figure 4, the expression levels of HSP70 were maintained during delivery and subsequent storage in the control group while increased at 5 h (*p* < 0.05) and 24 h (*p* < 0.05) of delivery in the treatment group in comparison with that at 0 h. The treatment group induced higher expression levels of HSP70 than the control group at 72 h (*p* < 0.05) of subsequent storage. The expression levels of HSP90 decreased in the control group at 5 h (*p* < 0.05) and 24 h (*p* < 0.05) of delivery, while they increased in the treatment group at 14 h (*p* < 0.01) and 24 h (*p* < 0.05) of delivery and continued to increase during subsequent storage at 48 h (*p* < 0.001), 72 h (*p* < 0.001) and 96 h (*p* < 0.001), in comparison with that at 0 h. The treatment group exhibited higher expression levels of HSP90 than that in the control group, especially at 5 h (*p* < 0.05), 14 h (*p* < 0.05) and 24 h (*p* < 0.001) of delivery and during subsequent storage for 48 h (*p* < 0.001), 72 h (*p* < 0.001) and 96 h (*p* < 0.05).

## 4. Discussion

In the porcine industry, artificial insemination (AI) is a commonly used and efficient tool for reproduction. Many factors could affect the results of AI, among which transport and storage of extended boar semen are receiving more and more attention [25]. Vibration and temperature changes are considered two main events that take place during transport, which exert a negative effect on boar sperm quality [3,26]. However, the vibration during transport shows no apparent influence on sperm quality when semen is kept at a constant temperature after collection, for example, during holding time at 22 °C or thereafter at 17 °C [26]. Therefore, the shipping temperature variation is the most fatal factor to boar semen transport, except for the case of long-lasting transport of semen on an extremely rough road surface. Due to the lack of evidence on the effect of shipping temperature changes on boar sperm quality and its underlying molecular changes, the present study investigated the effect of a commonly used semen shipping method, the Styrofoam box, on boar sperm quality during hot summer days by simulating the shipping practice in the laboratory. Our results demonstrated the negative impact exerted by temperature changes on boar sperm quality during boar semen shipping under hot environmental temperatures. The elevated temperature during transport promotes sperm apoptosis, which results in sperm quality loss. The results of the present study provide new evidence of the adverse effect of shipping temperature variation on sperm quality and theoretical information on the study of the mechanism of how temperature changes alter sperm quality and how sperm resists thermotic changes. Moreover, the results will help in improving the shipping method of boar semen in practice.

### 4.1. Effect of Temperature Changes during Delivery on Sperm Quality and Functionality

By using an animal model or scrotal insulation, many studies have demonstrated the adverse influence of heat stress on spermatogenesis, semen quality, and reproductive traits in bovine [27], mice [28], dogs [29], sheep [30], and boar [7] when exposed to heat stress environment. However, few studies focus on the effect of in vitro exposure of semen to heat stress on sperm quality and functionality. As it is well known, temperature management is crucial for semen handling and the subsequent semen storage and delivery. Boar sperm is susceptible to temperature changes [31], which could be attributed to the higher content of unsaturated fatty acid in the plasma membrane [32]. When sperm cells undergo temperature changes, lipid architecture and fluidity of sperm plasma membrane are altered [33,34], causing a loss in sperm quality and functionality. Previous evidence showed that storage of boar semen at temperatures > 20 °C or <15 °C results in sperm quality loss [35]. Thus, a storage temperature of 17–25 °C is recommended for boar semen for neither inducing cold shock nor excessive energy consumption [36]. Therefore, efforts have been made to keep the semen temperature in this range during both storage and delivery. Styrofoam boxes and car-loaded refrigerators are the two main methods used for semen delivery. Due to the high economic cost of the latter, a Styrofoam box is being widely used. However, shipping semen using a Styrofoam box can be greatly influenced by environmental temperatures, possibly resulting in semen temperatures higher than 20 °C on hot days or lower than 15 °C on cold days.

In order to explore the effect of temperature changes during semen delivery on hot summer days on boar sperm quality and functionality, the present study simulated regular semen delivery using a Styrofoam box under a constant temperature of 37 °C. Semen temperature was monitored, reaching up to 20 °C after 5 h of delivery and 37 °C (the environment temperature) after 24 h of delivery. Even though the Styrofoam box loaded with boar semen was well-sealed and packaged with ice bags inside, the semen temperature increased. A detrimental effect of temperature increase on boar sperm quality and functionality was observed. Sperm motility was markedly reduced after the temperature reached 30 °C, and it was impossible to recover sperm motility by subsequent storage back to 17 °C. The results imply that sperm motility loss caused by temperature changes during delivery is lethal and permanent. A decline in sperm viability and acrosome integrity was obvious after semen temperature reached 20 °C, which happened much earlier than that under constant 17 °C. Temperature changes during semen delivery under 37 °C deteriorated sperm quality, which could be explained by the maintenance of high ROS levels during the temperature-rising process. It has been reported that exposure of boar semen to 39 °C greatly decreases both total and progressive sperm motility while increasing ROS levels [6]. Furthermore, when the boar semen temperature is above 40 or 42 °C, sperm motion parameters decline rapidly to zero [7]. Exposure to higher temperatures shortens their lifespan owing to an insufficient energy supply and overproduction of ROS [5,6]. Moreover, the present study showed that sperm mitochondrial membrane potential manifested a similar pattern with storage time in semen stored under constant 17 °C and a delivery simulating model, being somehow higher in the latter one due to the increased metabolic activity induced by the elevated temperature. In this way, temperatures beyond the physiological range may cause more serious damage to sperm. As it is reported, the exposure of Holstein bull frozen–thawed sperm to 38.5~41 °C for 4 h leads to a loss in sperm motility and mitochondrial activity and an increase in ROS production and caspase activity [37]. Taken together, exposure of boar semen to elevated temperature may exert detrimental effects on boar sperm quality and functionality through ROS-mediated oxidative stress. Thus, temperature changes during delivery result in a loss in sperm quality and functionality, possibly leading to a decreased output of viable embryos and decreased field fertility, as demonstrated in the process of spermatogenesis [38]. To resolve this problem, mobile sensors for temperature and special extenders, e.g., supplemented with 1.0 mM L-arginine [6], that favor the maintenance of sperm quality could be helpful methods for semen transportation [21].

### 4.2. Effect of Temperature Changes during Delivery and Subsequent Storage (at 17 °C) on Sperm Apoptotic Levels

The compromised boar sperm quality during delivery on hot summer days could be attributed to cell apoptosis induced by temperature changes, which, together with autophagy, DNA damage and accumulation of ROS, are proved to be the main consequences of testicular heat stress [39]. However, whether in vitro exposure of boar semen to heat stress undergoes similar molecular changes remains unclear. In the present study, we focused on sperm apoptotic levels possibly induced by temperature variation. Our results confirmed that temperature changes during semen delivery under an environmental temperature of 37 °C obviously promoted the conversion of early apoptosis into late apoptosis in sperm. Meanwhile, levels of pro-apoptotic protein Bax and apoptosis effector Caspase 3 notably increased during delivery and subsequent storage at 17 °C up to 3 days, but the levels of anti-apoptotic protein Bcl-2 showed no obvious changes. In other words, the rising temperature of boar semen accelerates the sperm death process, which could be the consequence of hypoxia and increasing ROS production induced by the elevated temperature, as demonstrated in the case of testicular heat stress [27]. Excessive ROS cause lipid peroxidation in membranes and damage DNA, denaturing proteins and inducing apoptosis [40]. In turn, lipid peroxidation causes further ROS production, accelerating the apoptotic response [41]. Similar evidence has been reported when boars were exposed to an ambient temperature of 37–40 °C for 3 h per day during a week, resulting in apoptotic changes in boar sperm [42]. Notably, heat treatment at the body level increases the expression levels of Bcl-2 but induces no significant changes in Bax expression levels in boar sperm cells of different types in the testis [42], while no obvious changes in the expression levels of Bcl-2 and a significant increase in that of Bax were observed when boar semen was exposed to heat treatment in the present study. The different results obtained can probably be due to the difference in response to heat treatment between immature and mature boar sperm cells, being the latter one less vulnerable but less capable of adjusting themselves to the heat by regulating the ratio of Bcl-2 and Bax. When boar sperm are exposed to heat stress, whether in vitro or in vivo, apoptotic changes occur through an intrinsic pathway that involves mitochondria or an extrinsic pathway that involves Fas/FAS-ligand signaling through p53 activity [43], which may depend on the extent of damage induced by heat stress. The apoptotic intrinsic pathway involves a redistribution of Bax from its cytoplasm to a paranuclear location, up-regulation and phosphorylation of anti-apoptotic Bcl-2, cytosolic translocation of cytochrome c and DIABLO, activation of the initiator Caspase 9 and the executioner Caspases 3, 6, and 7, and cleavage of PARP [8]. This pathway has been demonstrated in heat-induced apoptosis in several male models [44]. Therefore, certain inhibitors of caspases and cytochrome c release can alleviate heat-induced germ cell apoptosis [45,46]. Taken together, the results of the present study confirmed the heat stress-induced apoptosis in boar sperm during semen delivery under high ambient temperatures.

### 4.3. Effect of Temperature Changes during Delivery and Subsequent Storage (at 17 °C) on Activation of Sperm AMPK and Expression Levels of HSP70 and HSP90

Several studies have demonstrated that AMPK activation promotes cell apoptosis or death under various stress conditions [47,48,49]. In boar sperm, AMPK was first found in the year of 2012, whose activation favors sperm quality and functionality [13,50]. In the present study, we observed increased AMPK phosphorylation at 5 h of simulating semen delivery (semen temperature increased to 20 °C) in relation to that of constant storage at 17 °C, which corresponds to the levels of both early and late sperm apoptosis at the same time point. Hereby, the activation of AMPK may be induced by temperature elevation via slightly increasing the expression levels of AMPK upstream factors LKB1 and CaMKK2. In addition, the significantly higher levels of ROS after 5 h of delivery may contribute to the activation of AMPK, as our previous study demonstrates that oxidative stress during liquid storage promotes AMPK activation of boar sperm [21]. Nonetheless, the levels of phosphorylated AMPK showed no difference in heat treatment and control groups at 14 h (semen temperature at 30 °C) and 24 h (semen temperature at 37 °C) but declined significantly after storing semen samples at 17 °C again, which could be attributed to the exhausting status of boar sperm that were severely damaged in terms of motion parameters and membrane integrity. As our results indicated, boar sperm quality was greatly altered after semen temperature reached 30 °C, which resulted in considerable incapability in ATP production. It seems that mild heat treatment (temperature increase from 17 °C to 20 °C) promotes AMPK activation, which induces sperm apoptosis. It is interesting that the activation of AMPK promoted sperm apoptosis under mild heat treatment in the present study rather than benefit the maintenance of sperm quality and functionality, as reported [13]. The controversial effect of activated AMPK could be due to its variation in regulating cell energetic homeostasis. Under the physiological range, AMPK activation plays a positive role in cell survival, while the opposite effect occurs when beyond the normal temperature range.

AMPK is associated with heat shock proteins under heat-stress conditions. Evidence shows that heat stress at the body level significantly increases the levels of serum adiponectin, AMPK, HSF, HSP27, HSP70, and HSP90 in dairy cows [51]. N-butyrate is partly involved in the microbiota-dependent intestinal expression of HSP70, acting through the HSF1 and AMPK pathways [52]. Furthermore, heat treatment increases HSP70 expression but decreases phosphorylated AMPK levels in the liver of rats. In contrast, 17-dimethylaminoethylamino-17-demethoxy-geldanamycin (17-DMAG) shows protective effects against heat stress in rats through the up-regulation of HSP70 and phosphorylated AMPK [53]. In addition, increased HSPs were observed in the sperm of men with varicocele and in those with oligozoospermia. An increased HSP90 was observed in oligozoospermia cases independent of varicocele [54]. A similar phenomenon was observed in the present study. Compared with constant storage at 17 °C, the expression levels of HSP70 and HSP90 in boar sperm significantly increased with time during delivery and subsequent storage at 17 °C, which is accompanied by an increase in phosphorylated AMPK when semen temperature reached 20 °C and an obvious decline in that when semen was subsequently stored at 17 °C after delivery. The current results indicate the beneficial role of AMPK activation against heat stress, which is linked with the increased expression of heat shock proteins.

HSP is categorized into six families in mammals, namely HSP100, HSP90, HSP70, HSP60, HSP40, and HSP27 [55]. Among those HSPs, HSP70 and HSP90 have been associated with boar sperm quality and fertility. HSP transcripts are rich in sperm, playing a vital role under stress conditions. The presence of HSP70 and HSP90 transcripts has been demonstrated in boar sperm. The transcript level of both HSPs shows a positive correlation with boar semen quality. Furthermore, both HSP proteins were detected in boar sperm [19]. A temperature increase from 19 °C to 25 °C leads to a decrease in HSP90 mRNA levels in boar ejaculates [56]. In addition, HSP70 levels in buffalo semen are higher in summer than in winter, which is positively correlated with sperm motility, live sperm count, and acrosome integrity. Positive associations of HSP70 and HSP90 levels in frozen–thawed buffalo semen with conception rate after AI has been reported [57]. The evidence reported implies that HSP70 and HSP90 show beneficial effects on sperm quality and fertility. However, the mechanisms of how they work require further study. Ikwebuge et al. [58] reported that HSPs protect cells from apoptosis damage and oxidative stress. HSP70 is localized in the equatorial segment of boar sperm, with a dynamic distribution when sperm experience capacitation and acrosome reaction [59]. When cells are heated, HSP70 increases SOD activity to protect them from oxidative damage. The decreased expression of HSP70 induced by a freezing–thawing procedure may explain the compromised sperm quality and functionality post-thaw [60]. In addition, HSP-70 reduces H_2_O_2_-induced ROS accumulation, TAK1/AMPK activation, and cardiomyocyte death [15]. HSP90 is localized in the midpiece of boar sperm, showing no distribution when sperm experience capacitation and acrosome reaction [61]. HSP90 maintains boar sperm motility and mitochondrial membrane potential when exposed to 40 °C for 24 h in a non-capacitating medium, while no effect of HSP90 exists in a capacitating medium [62]. Large molecular weight HSPs (HSP60, HSP70, HSP90, and HSP100) are adenosine triphosphate (ATP) dependent and act in various ways, including protein folding and translocation, cytoprotection, core hormone receptor control, and apoptosis regulation [63]. Similarly, the results of the present study showed that the increasing expression levels of HSP70 and HSP90, to some extent, correspond to the changes in the AMP/ATP ratio and the levels of sperm apoptosis during boar semen delivery and subsequent storage. The augmentation in HSP70 and 90 induced by temperature elevation during semen delivery could be involved in the mechanism of sperm resistance to heat stress. The activation of AMPK within the normal range could play a positive role in protecting boar sperm from heat-induced apoptosis, as the protective effect of 17-DMAG against heat stress in rats is demonstrated to act through up-regulating HSP70 and phosphorylating AMPK [53]. In addition, studies have reported that either Hsp90 or Hsp70 can bind and block the activation of apoptotic protease activating factor 1 (Apaf-1) and indirectly inhibit pro-caspase activation, apoptosis, as well as enhance abnormal cell survival [20]. The overexpression of HSP27, HSP60, HSP70, and HSP90 has proved to suppress apoptosis and prevent caspase activity of cells under stress [64]. It has been demonstrated that HSP90 and Cdc37 bind specifically to the kinase domain of LKB1. The association of HSP90 and Cdc37 with LKB1 regulates LKB1 stability and prevents its degradation by the proteasome [17]. Therefore, HSP70 and HSP90 may be involved in AMPK activation when sperm suffers heat stress by regulating the stability of AMPK upstream kinase LKB1, which favors sperm resistance to heat stress. On the other hand, HSP70 and HSP90 prevent sperm apoptosis through indirect inhibition of pro-caspase activation, which profits cell survival. Considering the association between HSPs and sperm quality and functionality, addition of combined HSPs (HSP60, HSP70, HSP90) into boar semen collected in summer was performed but no improvement in sperm quality or fertility was observed after storage at 17 °C for 48 h [65]. The biological functions of HSPs seem to be quite complicated. The inhibition of HSP90 of boar sperm under capacitation and non-capacitation conditions shows different effects on sperm motility [62]. The evidence indicates the variation in the regulatory role of HSPs in sperm quality and fertility under different environmental conditions, biological processes, and signaling pathways. More effort is needed to obtain a full understanding of that.

## 5. Conclusions

In conclusion, temperature elevation during boar semen delivery promotes sperm apoptosis, resulting in damage to sperm quality and functionality. HSP70 and HSP90 could be involved in AMPK activation and anti-apoptotic processes under increased semen temperatures, which favor the survival of boar sperm.

## Figures and Tables

**Figure 1 animals-13-03203-f001:**
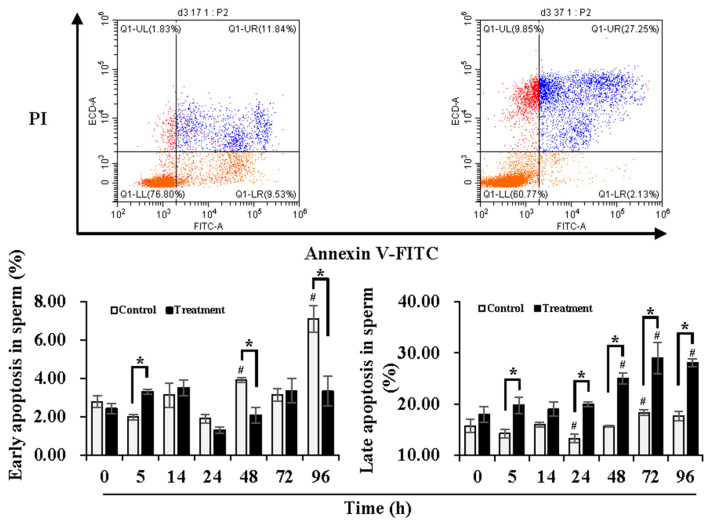
Effects of temperature changes during delivery and subsequent storage (at 17 °C) on the apoptotic levels of boar sperm. Dot plots show staining with Annexin V-FITC and PI in boar sperm after storage at 17 °C for 96 h. The lower right quadrant indicates levels of early apoptosis (sperm population with negative PI and positive Annexin-V). The upper right quadrant indicates levels of late apoptosis (sperm population with positive PI and positive Annexin-V). Histogram displays the levels of sperm early apoptosis (**left**) and late apoptosis (**right**). Bars represent mean ± SEM. # Indicates significant differences in comparison with values at 0 h, *p* < 0.05; * Indicates significant differences between control and treatment groups, *p* < 0.05.

**Figure 2 animals-13-03203-f002:**
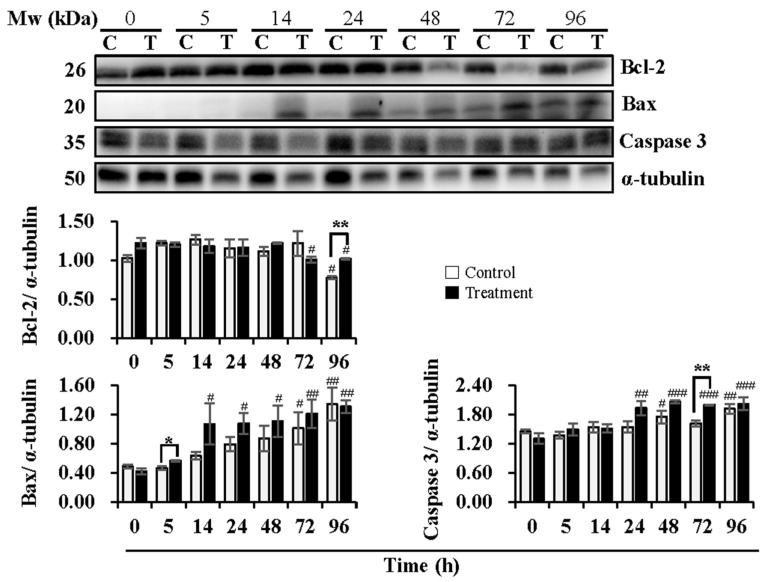
Effects of temperature changes during 24 h of delivery and subsequent storage (at 17 °C) for 3 days on the levels of Bcl-2, Bax and Caspase 3 in boar sperm. Boar semen samples of 17 °C were kept at a constant temperature of 17 °C (Control) and a higher ambient temperature of 37 °C (Treatment). All Western blots were representative of 3 biological replicates. Bar graphs stand for data expressed as mean ± SEM. # Indicates significant differences in comparison with values at 0 h, *p* < 0.05; ##, *p* < 0.01; ###, *p* < 0.001. * Indicates significant differences between control and treatment groups, *p* < 0.05; **, *p* < 0.01.

**Figure 3 animals-13-03203-f003:**
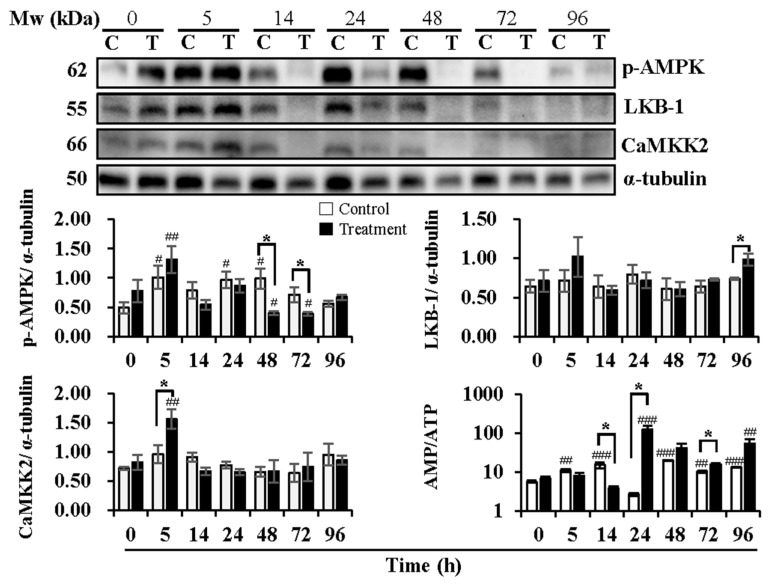
Effects of temperature changes during 24 h of delivery and subsequent storage (at 17 °C) for 3 days on the levels of phosphorylated AMPK, levels of AMPK upstream kinase CaMKK2 and LKB1, and intracellular AMP/ATP ratio in boar sperm. Boar semen samples of 17 °C were kept at a constant temperature of 17 °C (Control) and a higher ambient temperature of 37 °C (Treatment). All Western blots were representative of 3 biological replicates. Bar graphs stand for data expressed as mean ± SEM. # Indicates significant differences in comparison with values at 0 h, *p* < 0.05; ##, *p* < 0.01; ###, *p* < 0.001. * Indicates significant differences between control and treatment groups, *p* < 0.05.

**Figure 4 animals-13-03203-f004:**
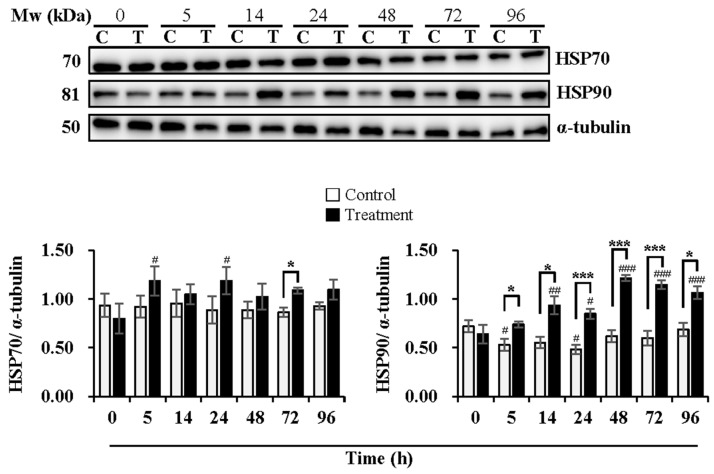
Effects of temperature changes during 24 h of delivery and subsequent storage (at 17 °C) for 3 days on the levels of HSP70 and HSP90 in boar sperm. Boar semen samples of 17 °C were kept at a constant temperature of 17 °C (Control) and a higher ambient temperature of 37 °C (Treatment). All Western blots were representative of 3 biological replicates. Bar graphs stand for data expressed as mean ± SEM. # Indicates significant differences in comparison with values at 0 h, *p* < 0.05; ##, *p* < 0.01; ###, *p* < 0.001. * Indicates significant differences between control and treatment groups, *p* < 0.05; ***, *p* < 0.001.

**Table 1 animals-13-03203-t001:** Effects of temperature changes during delivery and subsequent storage (at 17 °C) on total sperm and progressive motility.

		Total Motility, %		Progressive Motility, %	
		Control	Treatment	Control	Treatment
During delivery	0 h	80.00 ± 5.64 ^a^	74.33 ± 8.09 ^a^	45.00 ± 5.86 ^a^	42.00 ± 5.51 ^a^
	5 h	81.33 ± 8.01 ^a^	77.33 ± 6.96 ^a^	45.33 ± 7.54 ^a^	41.67 ± 5.36 ^a^
	14 h	86.33 ± 3.48 ^a^	80.33 ± 9.70 ^a^	50.33 ± 2.03 ^a^	53.33 ± 6.69 ^a^
	24 h	85.33 ± 5.90 ^ax^	5.00 ± 3.22 ^by^	49.00 ± 3.61 ^ax^	0.67 ± 0.67 ^by^
Subsequent storage at 17 °C	48 h	85.00 ± 7.21 ^ax^	3.33 ± 1.76 ^by^	47.67 ± 7.84 ^ax^	0.33 ± 0.33 ^by^
	72 h	46.33 ± 5.04 ^bx^	2.33 ± 1.16 ^by^	26.00 ± 4.04 ^bx^	0.33 ± 0.33 ^by^
	96 h	9.00 ± 3.22 ^c^	2.00 ± 1.00 ^b^	2.67 ± 1.45 ^c^	0.00 ± 0.00 ^b^

Note: To simulate semen delivery under hot summer days, one aliquot of each semen sample at 17 °C was wrapped with thermal insulation film and placed in a Styrofoam box together with an ice pack. The Styrofoam box was placed in an incubator at 37 °C for 24 h. Semen temperature reached to 17 °C, 20 °C, 30 °C and 37 °C when stored for 0 h, 5 h, 14 h and 24 h, respectively. Constant storage at 17 °C was taken as control. letters a–c indicate significant differences in sperm parameters between storage times (same column), *p* < 0.05; x and y indicate significant differences in sperm parameters between control and treatment groups (same row), *p* < 0.05. The same as below.

**Table 2 animals-13-03203-t002:** Effects of temperature changes during delivery on sperm viability, acrosome integrity, intracellular ROS production and mitochondrial membrane potential.

Sperm Parameters	Delivery Time				
		0 h	5 h	14 h	24 h
Viability, %	Control	72.17 ± 0.48 ^a^	64.08 ± 1.03 ^a^	74.28 ± 0.98 ^bx^	62.49 ± 1.43 ^bx^
	Treatment	69.44 ± 1.45 ^a^	62.26 ± 3.17 ^b^	58.06 ± 2.09 ^by^	48.29 ± 1.19 ^by^
Percentage of viable sperm with damaged acrosome, %	Control	0.07 ± 0.02 ^a^	0.15 ± 0.04 ^ab^	0.19 ± 0.04 ^bx^	0.16 ± 0.03 ^ab^
	Treatment	0.11 ± 0.02 ^a^	0.12 ± 0.03 ^a^	0.35 ± 0.03 ^by^	0.28 ± 0.07 ^b^
Intracellular ROS production, 10^6^ fluorescence units/10^6^ viable sperm	Control	5.97 ± 0.47 ^ax^	3.96 ± 0.22 ^b^	2.54 ± 0.09 ^cx^	2.88 ± 0.32 ^bc^
	Treatment	4.42 ± 0.39 ^y^	3.41 ± 0.10	4.02 ± 0.56 ^y^	3.42 ± 0.45
Mitochondrial membrane potential of viable sperm, %	Control	53.10 ± 4.90 ^a^	60.02 ± 5.17 ^ab^	63.92 ± 2.52 ^ab^	68.44 ± 2.81 ^b^
	Treatment	55.35 ± 2.27 ^a^	52.79 ± 8.37 ^ab^	68.91 ± 2.24 ^ab^	74.98 ± 0.82 ^b^

Note: To simulate semen delivery under hot summer days, one aliquot of each semen sample at 17 °C was wrapped with thermal insulation film and placed in a Styrofoam box together with an ice pack. The Styrofoam box was placed in an incubator at 37 °C for 24 h. Semen temperature reached to 17 °C, 20 °C, 30 °C and 37 °C when stored for 0 h, 5 h, 14 h and 24 h, respectively. Constant storage at 17 °C was taken as control. letters a–c indicate significant differences in sperm parameters between storage times (same column), *p* < 0.05; x and y indicate significant differences in sperm parameters between control and treatment groups (same row), *p* < 0.05.

## Data Availability

The data presented in this study is available in the article.

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
