# Peer review of "Temperature Elevation during Semen Delivery Deteriorates Boar Sperm Quality by Promoting Apoptosis"

_animals, 2023, doi:10.3390/ani13203203_

Round 1

Reviewer 1 Report

I have only comments on the materials and method  as follows:

Please describe the process of fresh semen dilution, whether the author diluted or not-diluted fresh semen before sending with styrofoam box. Although, the author mentioned on semen extender in the last paragraph of this part. Please described this clearly at the beginning of method.

Author Response

Point 1. I have only comments on the materials and method as follows:

Please describe the process of fresh semen dilution, whether the author diluted or not-diluted fresh semen before sending with styrofoam box. Although, the author mentioned on semen extender in the last paragraph of this part. Please described this clearly at the beginning of method.

Response 1: The authors appreciate your valuable advice. We agree that a clear statement of detailed semen handling at the beginning will definitely help to provide a clear idea of the experiment. Actually, after semen collection, ejaculates were diluted to a concentration for AI using a commercial extender and packaged following the conventional practice set in the AI station. Thereafter, semen doses were delivered to the lab for further experiments. The detailed information has been added in the Materials and Methods.

Reviewer 2 Report

The authors have done a decent job of creating a robust dataset investigating a production problem at a molecular level to understand how temperature impacts semen quality. 

General comments:

It would be beneficial for more details on the semen preparation. What was the volume of the pooled samples created? With multiple samples collected, why did the authors only create one pool so as to not interrupt temperature exposure during sampling? How much of the original pool was left leading into the last few samples? Could this have impacted results?

Specific comments:

Line 78: switch summer and hot, it should read hot summer days

Line 133: kept is misspelled

Line 147: remove the hyphen in the word minimum

Line 168: remove the hyphen in the word incubated

Line 447: switch summer and hot, it should read hot summer days

Line 483: switch summer and hot, it should read hot summer days

There are multiple times throughout the manuscript when the verb placement is incorrect or word order is outside of the correct sentence structure. 

Author Response

The authors have done a decent job of creating a robust dataset investigating a production problem at a molecular level to understand how temperature impacts semen quality.

Point 1. General comments:

It would be beneficial for more details on the semen preparation. What was the volume of the pooled samples created? With multiple samples collected, why did the authors only create one pool so as to not interrupt temperature exposure during sampling? How much of the original pool was left leading into the last few samples? Could this have impacted results?

Response 1: Thank you for your valuable comments. The authors agree that more details for semen preparation will help in a better understanding of this study. We have added the related information, briefly, ten semen doses of 80 mL for each time were used for experiments, from each of which 10 mL were taken out to create a pooled sample of 100 mL. The reason why we just used 10 ejaculates to create 1 pooled sample is to avoid variations between boars and ejaculates, and to have a better control of temperature exposure for each time. For the last question, actually it was far from enough with the pooled semen samples of 100 mL for all the measurements and protein analysis. The pooled semen samples were aliquoted into several parts according to measurement timepoints. So for each timepoint, the amount of semen samples for assays was enough and the temperature exposure was well managed. In this case, we believe that there was hardly impact of sample volume on the results.

Specific comments:

Point 2. Line 78: switch summer and hot, it should read hot summer days

Line 133: kept is misspelled

Line 147: remove the hyphen in the word minimum

Line 168: remove the hyphen in the word incubated

Line 447: switch summer and hot, it should read hot summer days

Line 483: switch summer and hot, it should read hot summer days

Response 2: The authors are really grateful for your kind remind on those mistakes or errors in the text. All the suggestions have been taken and modifications have been made accordingly. Besides, the authors re-checked closely those sentences.

Reviewer 3 Report

Material and method

 What type of extender was used for the ejaculates transport? Because 30 hours of transport plus the 24 hours at 37ºC and the 96 hours at 17ºC results in 6.5 days, too much. The description of the extender is relevant. There are short term extender, long term extender, extra long term… and the election has influence on the results.

The paper talks about the ACROMAX PLUS extender (ZoitechLab), used for the controls, but  after centrifugation semen was washed  with BTS. What was the extender used for the dosis?

What is the age of the males? The response to Tª is not the same whether they are young or adults, as mentioned in the work itself.

The number of simples, ejaculates and replicates it´s not clear:

Line 39: Three pooled semen samples from 10 Duroc boars (3 ejaculates per boar) were used in this study.

Line 113: Thirty ejaculates from 10 Duroc boars (3 ejaculates per boar) were used in this study

Line 123: Three replicates were performed. For each replicate, ten ejaculates were mixed to create one pooled semen sample

Line 129:  Semen samples stored for 0 h, 5 h, 14 h and 24 h and their temperature reached to 17 °C, 20 °C, 130 30 °C and 37 °C, respectively. The temperatures reached will be results, not material al methods.

Line 147: min-imum of 400 sperm per sample. Remove hyphen

Line 168: incu-bated at 37. Remove hyphen

The way in which the concentration is adjusted and the extender used to do it is not defined neither for motility, intracellular ATP, ADP, and AMP content or Western blotting.

Line 144: 5 μL semen sample (25× 106 cells/mL)

Line195:  sperm samples of 100 μL (20 × 106 cells/mL) were mixed  

Line 220: Samples containing 1.2× 108 sperm cells were centrifuged at 2,400 × g for 3 min at 4ºC

Results

Line 266: The first part of the note is not necessary, it is included in material and method: “To simulate semen delivery under hot summer days, one aliquot of each semen sample at 17 266 °C was wrapped with thermal insulation film and placed in a Styrofoam box together with an ice 267 pack. The Styrofoam box was placed in an incubator at 37 °C for 24 h. Semen temperature reached 268 to 17 °C, 20 °C, 30 °C and 37 °C when stored for 0 h, 5 h, 14 h and 24 h, respectively. Constant storage 269 at 17 °C was taken as control”.

Discussion

Information that is already reflected in the material and methods and in the introduction is repeated, especially in section 4.1.

In general, the discussion is not such, it is too generic, and no real reason is given why temperature affects sperm or how it does so.

Line 469:  Why temperature beyond physiological range may cause more serious damage to sperm than treatment id this one is more stressing?

Line 489:  temperature of 37 °C obviously promoted the conversion of early apoptosis into late apoptosis in sperm. Why does the temperature effect take longer to take effect in this case?

Conclusions

The aim of the paper is to investigate the underlying molecular responses of sperm to elevated temperature rise during a common semen delivery method on boar sperm quality and functionality. So, the next affirmation it´s not a conclusion because is not related with the objective:

“With the increasing intensification and standardization for boar studs in the future, there must be a great demand in optimizing semen transport practice. Temperature control and development of semen extender with special properties could be the promising solutions”

This would be a reflection that we extract from the results to try to improve transport protocols, but it is not a conclusion obtained from the results of the work.

Author Response

Material and method

Point 1. What type of extender was used for the ejaculates transport? Because 30 hours of transport plus the 24 hours at 37ºC and the 96 hours at 17ºC results in 6.5 days, too much. The description of the extender is relevant. There are short term extender, long term extender, extra long term… and the election has influence on the results.

Response 1: Thank you for your comments. The extender used for diluting ejaculates before packaging and transport was a long-term type for 7-10 days but the detailed information is not accessible from the boar station. The authors added “long-term extender was used for dilution of ejaculates” in the manuscript according to your suggestions. We agree with you that the type of extender really matters for its effect on preservation time and efficiency of boar semen. Therefore, we followed the practice in boar stud for the conventional production and transport of semen.

Point 2. The paper talks about the ACROMAX PLUS extender (ZoitechLab), used for the controls, but after centrifugation semen was washed with BTS. What was the extender used for the dosis?

Response 2: Actually, the extender ACROMAX PLUS was used for further dilution before quality assays, not only for controls. The sperm samples obtained by centrifugation were washed with PBS. We have put all the details in the text. Concerning the extender for doses, as we explained in Point 1, before packaging ejaculates were diluted with a long-term commercial extender.

Point 3. What is the age of the males? The response to Tª is not the same whether they are young or adults, as mentioned in the work itself.

Response 3: Thank you for your remind on this information. We have added this to the text. At present, the rate of replacement for boars is very high. The boars used for conventional semen production are of 1-2 years old. To avoid the variations between boar individuals and between ejaculates, we used 30 ejaculates from 30 boars (one ejaculate for one boar) in this study.

Point 4. The number of simples, ejaculates and replicates it´s not clear:

Line 39: Three pooled semen samples from 10 Duroc boars (3 ejaculates per boar) were used in this study.

Line 113: Thirty ejaculates from 10 Duroc boars (3 ejaculates per boar) were used in this study

Line 123: Three replicates were performed. For each replicate, ten ejaculates were mixed to create one pooled semen sample

Response 4: In the abstract, the authors described the creation of 3 pooled samples to give the idea of the experiment design. In the materials and methods, those information were consistent with that in the abstract. In the beginning, we presented the total number of ejaculates and boars used for semen doses and depicted replicates information at the end of this part. Briefly, we collected 1 ejaculate from 10 boars for each replicate and created 1 pooled semen sample for experiment. Totally, 3 replicated were performed.

Point 5. Line 129:  Semen samples stored for 0 h, 5 h, 14 h and 24 h and their temperature reached to 17 °C, 20 °C, 130 30 °C and 37 °C, respectively. The temperatures reached will be results, not material al methods.

Response 5: Thank you for your comments. The authors were thinking that the temperature changes with delivery time could be results or part of the methods. With an aim of giving a clear idea of the experiment design, we preferred to put the information of temperatures of different timepoints in the section of materials and methods.

Point 6. Line 147: min-imum of 400 sperm per sample. Remove hyphen

Line 168: incu-bated at 37. Remove hyphen

Response 6: The mentioned errors have been corrected.

Point 7. The way in which the concentration is adjusted and the extender used to do it is not defined neither for motility, intracellular ATP, ADP, and AMP content or Western blotting.

Line 144: 5 μL semen sample (25× 106 cells/mL)

Line195:  sperm samples of 100 μL (20 × 106 cells/mL) were mixed 

Line 220: Samples containing 1.2× 108 sperm cells were centrifuged at 2,400 × g for 3 min at 4ºC

Response 7: Yes. The concentration or amount of semen samples for quality assays, ATP test and Western blotting is not the same. And it is not necessary to adjust the concentration of semen samples in all these methods used. For motility assay, we set the concentration as 25× 106 cells/mL to get a better picture in the CASA system for analysis. For ATP test, the concentration was set according to the literature and kit instructions. For WB analysis, the concentration was set after a long-term trial in our laboratory to get the best results. Furthermore, the most important thing for this study is that all the assays were conducted on a basis of same experiment conditions, which made it possible to evaluate the effect of temperature exposure.

Results

Point 8. Line 266: The first part of the note is not necessary, it is included in material and method: “To simulate semen delivery under hot summer days, one aliquot of each semen sample at 17 266 °C was wrapped with thermal insulation film and placed in a Styrofoam box together with an ice 267 pack. The Styrofoam box was placed in an incubator at 37 °C for 24 h. Semen temperature reached 268 to 17 °C, 20 °C, 30 °C and 37 °C when stored for 0 h, 5 h, 14 h and 24 h, respectively. Constant storage 269 at 17 °C was taken as control”.

Response 8: We agree with you. Here we put the information of experiment design in order to express the results in a logic way and may provide better understanding of the results.

Discussion

Point 9. Information that is already reflected in the material and methods and in the introduction is repeated, especially in section 4.1.

In general, the discussion is not such, it is too generic, and no real reason is given why temperature affects sperm or how it does so.

Response 9: The repetition here was to make all the contents a whole complete story. The authors agree that there are still many more studies needed to do for a thorough understanding of the effect of temperature changes on sperm quality and functionality. Our findings solely demonstrated the detrimental effect of temperature and associations between temperature changes, sperm quality, AMPK, apoptosis, and heat shock proteins.

Point 10. Line 469:  Why temperature beyond physiological range may cause more serious damage to sperm than treatment id this one is more stressing?

Response 10: The sentence “In this way, temperature beyond physiological range may cause more serious damage to sperm.” was to express that temperature more than 17℃ may cause more severe damage to sperm as detrimental effect has been already observed at 17℃. In the following sentence, we discussed about the temperatures much higher and their influence on sperm quality and functionality. According to the results of literature, yes, temperature beyond physiological range induces oxidative stress and apoptosis, which leads to shorter lifespan of sperm.

Point 11. Line 489:  temperature of 37 °C obviously promoted the conversion of early apoptosis into late apoptosis in sperm. Why does the temperature effect take longer to take effect in this case?

Response 11: In this study, we took questions from the conventional semen production, temperature changes during delivery on sperm quality. The experiment condition was that Styrofoam box, the conventional method for semen transport, was placed in an incubator of 37℃ to simulate semen delivery in hot summer days. As we presented in the experiment design, the temperature was rising from 17℃ to 20℃ after 5 h, to 30℃ after 14 h, to 37℃ after 24 h. Thus, it indeed took time to reach to a certain temperature, for example above 30℃, to display the detrimental effect of temperature changes on sperm quality.

Conclusions

Point 12. The aim of the paper is to investigate the underlying molecular responses of sperm to elevated temperature rise during a common semen delivery method on boar sperm quality and functionality. So, the next affirmation it´s not a conclusion because is not related with the objective:

“With the increasing intensification and standardization for boar studs in the future, there must be a great demand in optimizing semen transport practice. Temperature control and development of semen extender with special properties could be the promising solutions”

This would be a reflection that we extract from the results to try to improve transport protocols, but it is not a conclusion obtained from the results of the work.

Response 12: The authors agree with you on the expression of the conclusion. We deleted the sentences you mentioned.